# Hybrid Passivated Red Organic LEDs with Prolonged Operation and Storage Lifetime

**DOI:** 10.3390/molecules27092607

**Published:** 2022-04-19

**Authors:** Dan-Dan Feng, Shuang-Qiao Sun, Wei He, Jun Wang, Xiao-Bo Shi, Man-Keung Fung

**Affiliations:** 1Jiangsu Key Laboratory for Carbon-Based Functional Materials & Devices, Institute of Functional Nano & Soft Materials (FUNSOM), Soochow University, Suzhou 215123, China; change071125@163.com (D.-D.F.); 20214014016@stu.suda.edu.cn (S.-Q.S.); 20204014026@stu.suda.edu.cn (W.H.); 2Institute of Organic Optoelectronics, Jiangsu Industrial Technology Research Institute (JITRI), Suzhou 215123, China; wangj@jitriioo.com (J.W.); shixb@jitriioo.com (X.-B.S.); 3Macao Institute of Materials Science and Engineering (MIMSE), MUST-SUDA Joint Research Center for Advanced Functional Materials, Zhuhai MUST Science and Technology Research Institute, Macau University of Science and Technology, Macau 999078, China

**Keywords:** red organic LEDs, hybrid encapsulation, Si_x_N_y_, operation lifetime, storage lifespan

## Abstract

In addition to mobile and TV displays, there is a trend of organic LEDs being applied in niche markets, such as microdisplays, automobile taillights, and photobiomodulation therapy. These applications mostly do not require to be flexible in form but need to have long operation lifetimes and storage lifespans. Using traditional glass encapsulation may not be able to fulfill the rigorous product specification, and a hybrid encapsulation method by combining glass and thin-film encapsulation will be the solution. Conventional thin-film encapsulation technology generally involves organic and inorganic multilayer films that are thick and have considerable stress. As a result, when subjected to extreme heat and stress, the film easily peels off. Herein, the water vapor transmission rate (WVTR) of a 2 µm silicon nitride film prepared at 85 °C is less than 5 × 10^−5^ g/m^2^/day and its stress is optimized to be 23 MPa. Red organic LEDs are passivated with the hybrid encapsulation, and the *T_95_* lifetime reaches nearly 10 years if the LED is continuously driven at an initial luminance of 1000 cd/m^2^. In addition, a storage lifespan of over 17 years is achieved.

## 1. Introduction

With the rise of the Internet of Things, organic LEDs [1,2,3,4,5] have been gradually attracting attention in wearable electronics [6,7,8]; mobile and interactive displays and microdisplays [9,10,11,12,13,14]; photodynamic therapy [15,16,17,18,19,20]; and automotive taillights [21,22,23] because of their appealing characteristics, such as thinness and lightness, fast response time, low power consumption, and low heat generation. Some of the specific applications require extremely high stability.

OLEDs in vehicle taillights will be subjected to high temperatures and humidity, while the devices need to maintain high brightness. Therefore, it is indispensable to have a reliable passivation barrier that completely protects water-sensitive OLED devices from moisture molecules. Among various materials that exhibit good barrier properties, AlO_x_ has been commonly used as an encapsulation material because of its good compatibility with the atomic layer deposition (ALD) process and high moisture impermeability [24,25,26,27,28]. Since the ALD method requires a relatively longer process time than other deposition methods, such as plasma-enhanced chemical vapor deposition (PECVD), inkjet printing, and magnetron sputtering, AlO_x_ prepared by ALD should be much thinner, resulting in poor coverage of the oxide layer on OLEDs and hence a degraded encapsulation performance. To solve the above problems, many have been trying to improve the properties of the inorganic layer. Chen et al. used CeO_2_ and polydivinylbenzene for encapsulation at room temperature. However, the barrier had a water vapor transmission rate of 1.81 g/m^2^/day at 30 °C and 100% relative humidity (RH) [29], which may not be enough to meet the encapsulation requirements. In addition, thin-film encapsulation technology generally involves organic and inorganic multilayer films that are thick and have considerable stress. As a result, when subjected to extreme heat and stress, the film easily peels off and moisture is more prone to infiltrate the device, leading to the formation of dark spots.

In this paper, a simple and reliable thin-film encapsulation barrier with a bilayer structure (silicon nitride (Si_x_N_y_) and Al) is proposed. Si_x_N_y_ and Al are well known to be outstanding passivation films. There are two reasons to adopt Al. Firstly, sputtered Al film has tensile stress, which can compensate the compressive stress of Si_x_N_y_. Secondly, Al can enhance the optical reflectivity because a high-reflective-surface OLED taillight is generally required by the car industry. Furthermore, we combine this thin-film encapsulation method with glass encapsulation, which not only reduces stress to less than 25 MPa but also increases the operation lifetime and the storage lifespan of red-light OLEDs by 1.2- and 1.9-fold, respectively, and the EQE can still remain at 26% when the device brightness reaches 3000 cd/m^2^.

## 2. Experimental Section

### 2.1. Optimization of Encapsulation Layers

To prepare Si_x_N_y_ films by PECVD, we used silane (SiH_4_), hydrogen (H_2_), ammonia (NH_3_), and nitrogen (N_2_) at flow rates of 20, 20, 40, and 200 standard cubic centimeters per minute (sccm). During the deposition process, the chamber pressure was maintained at 1.4 Torr and the RF power was set to 200 W. The deposition temperatures of the Si_x_N_y_ films were 55, 65, 75, 85, and 95 °C. Once the temperature condition was optimized, the thicknesses of the Si_x_N_y_ films were controlled to be 0.5, 1, 1.5, 2, and 2.5 µm so as to further optimize the water vapor transmission rate (WVTR), stress, the transmittance, and the refractive index. For Al deposition, the inert gas Ar at 50 sccm was introduced. Its pressure was controlled to 2 Pa; the RF power was 200 W; and aluminum films of 100, 200, 300, 400, and 500 nm were deposited.

### 2.2. Device Fabrication

Commercially purchased indium tin oxide (ITO) glass substrates were ultrasonically treated for 10 min each with acetone, ethanol, and deionized water; baked for 2 h at 100 °C; and treated with ozone for 15 min before use. The ITO was used as the anode, which had a thickness of 135 nm and a sheet resistance of 15 Ω/□. The device was prepared in a thermal evaporator with a working pressure of less than 4 × 10^−6^ Torr. All the devices in the present study had the same structure, consisting of hole-injection layer, a hole-transport layer, an emissive layer composed of a phosphorescent emitter with 2% vol.% doped into the host, an electron transport layer, an electron injection layer using 8-hydroxyquinolinolato-lithium (Liq), and a cathode using Al. Following evaporation, the devices were packaged using various methods. All encapsulation was performed in a laboratory environment without precise control of particles and moisture.

### 2.3. Testing of Encapsulation Layers and Devices

The film stress was measured by a Frontier Semiconductor (FSM) 500TC instrument. The WVTR was measured by a MOCON water vapor transmittance tester (Permatran-W3/34G). ZEISS G500 was used for capturing SEM images of the silicon nitride film. The aging life was measured by a ZJLS-4 OLED aging tester. The current-efficiency-versus-current-density (CE–J) and luminance-versus-current-density (L–J) characteristics of the devices were measured using a Keithley 2400 source. All device characterizations were carried out at room temperature under ambient conditions.

## 3. Results and Discussions

### 3.1. Characteristics of Si_x_N_y_ Thin Films Deposited by PECVD

Intrusion of water and oxygen vapor will severely shorten the lifetime of an OLED. The primary purpose of encapsulation is to isolate water and oxygen from the ambient [30]. Since a Si_x_N_y_ thin film has outstanding mechanical and electrical features, such as low porosity, high density and high dielectric constant, we used it as an insulation and passivation layers in OLEDs. To avoid the degradation of organic materials at high temperatures [31], the preparation of the encapsulation layer should be a low-temperature process. However, due to the large internal stress of Si_x_N_y_ films prepared at low temperatures, the film may delaminate and crack, resulting in device failure and poorer reliability. It is, therefore, necessary to measure the residual stress of the films.

We first changed the substrate deposition temperature to 55, 65, 75, 80, 85, 90, and 95 °C and then deposited a 2-µm-thick Si_x_N_y_ thin film. Their film stress values (Figure 1a) were −178.3, −88.5, −47.3, −35.0, −23.2, −28.4, and −30.1 MPa, respectively (a negative value indicates compressive stress, while a positive value indicates tensile stress). The residual stress will directly affect the stability and reliability of the passivation layer. We can see that the stress values between 80 °C and 95 °C are similar. Since organic materials may suffer damage at a high temperature, the optimized deposition temperature was then selected to be 85 °C. Next, we changed the thickness of the silicon nitride film to 0.5, 1, 1.5, 2, and 2.5 µm at the optimized temperature of 85 °C, resulting in WVTRs of 1.2 × 10^−3^, 1.7 × 10^−4^, 6 × 10^−5^, 5 × 10^−5^, and 5 × 10^−5^ g/m^2^/day, respectively (Figure 1b). It should be noted that the WVTR was high in the beginning because the sensors of the MOCON WVTR tester were in contact with air prior to the loading of our samples. After our samples were installed and the instrument was entirely enclosed, it took time for the sensors to stabilize and the WVTR then dropped to a steady state. The steady state reflects the actual permeation rate of the water vapor penetrating into the thin film, so all the WVTR values were obtained after 5 days of testing. We can also see that when 2-μm- and 2.5-μm-thick Si_x_N_y_ films were deposited, the measured WVTR value reached the detection limit of the MOCON tester. In addition, the measured residual stress values became −25.7, −12.6, −19.7, −23.2, and −46.4 MPa, respectively (Figure 1c).

Considering the WVTR and stress, a 2-µm-thick Si_x_N_y_ film was finally selected as the encapsulation layer. Appendix A shows a scanning electron microscopic cross-sectional image of a 2-µm-thick Si_x_N_y_ film. It can be seen in Figure 1c that the stress for the 0.5-μm-thick film was relatively large, which may be attributed to the unstable deposition condition during the initial deposition stage. Furthermore, sputtered aluminum has a tensile stress, which may compensate for the compressive stress of Si_x_N_y_. The residual stress of aluminum films with thicknesses of 100, 200, 300, 400, and 500 nm prepared by magnetron sputtering was 5.4, 10.3, 21.1, 45.1, and 57.2 MPa, respectively (Figure 1d). It is then predicted that a composite film of Si_x_N_y_ (2 µm)/Al (300 nm) may have the corresponding compressive and tensile stress effectively cancelling each other. The Si_x_N_y_ (2 µm)/Al (300 nm) stress was also measured experimentally, which was only −1.0 MPa (Appendix A).

### 3.2. Encapsulation Performance of Red OLEDs

The aforementioned passivation films were applied to OLED encapsulation. We fabricated phosphorescent red-emitting OLEDs with an architecture as shown in Figure 2a. Four different encapsulation methods were used. They were a bare glass cover with desiccant film (Device R1); 2-μm-thick Si_x_N_y_ (Device R2); 2-μm-thick Si_x_N_y_ with a glass cover (Device R3); and 2-μm-thick Si_x_N_y_, 300 nm aluminum film with a glass cover (Device R4). It can be seen in Figure 3 that both the Si_x_N_y_ and Si_x_N_y_/Al encapsulation methods have no influence on the device performance and the electroluminescence (EL) spectrum when compared to typical glass-cover encapsulation. All the red OELDs have maximum external quantum efficiencies (EQEs) of over 30%, and the EQE can remain at 26% when the device brightness reaches 3000 cd/m^2^. The reduced efficiency roll-off is attributed to the low dopant concentration (2% vol.%) used in the emitter, thus diminishing its concentration quenching and charge trapping effect, as can be commonly found in typical phosphorescent OLEDs.

In addition, we investigated the *T_95_* lifetime of the red OLED driven at a high current density of 50 mA/cm^2^ (*T_95_*: the time for brightness to decay to 95 percent of its initial brightness at 50 mA/cm^2^), in which the *T_95_* lifetime for R1, R2, R3, and R4 was 460, 240, 530, and 560 h, respectively (Figure 4). We also carried out the lifetime measurements for a device encapsulated with 1-μm-thick Si_x_N_y_, and the *T_95_* lifetime was 160 h, as seen in Appendix A. By converting the initial brightness into 1000 *cd/m*^2^ using Equation (1) [32], the *T_95_* lifetime of R1–R4 became 63,800; 31,700; 76,300; and 84,500 h, respectively (Appendix A).
(1)$$T_{95}=T_{0}*[{\left(\frac{L_{0}}{1000\;cd/m^{2}}\right)}^{n}]$$
where *L_0_* is the initial luminance of the device tested at a current density of 50 mA/cm^2^, *n* is an acceleration factor in which *n* = 1.5 is commonly used, and *T_0_* is the time for brightness to decay to 95 percent of its initial brightness at 50 mA/cm^2^. It is realized that the single-film hybrid encapsulation (R3) outperforms the conventional encapsulation (R1) by 20% in its lifetime, while the double-layer hybrid encapsulation (R4) surpasses R1 by 32%.

To further examine the encapsulation performance of Si_x_N_y_ film, we evaluated the storage lifespan of the R1–R4 devices by a high-temperature/high-humidity test (at conditions of 85 °C and 85% relative humidity, the so-called 85/85 test). The device encapsulated solely with Si_x_N_y_ (R2) has the shortest storage lifetime, of 45 h. The shrinkage of the encapsulation film under a high temperature is the cause for the device failure. To improve it, we put a glass cover on top of the 2-μm-thick Si_x_N_y_ film (R3) to firmly press the encapsulation film and diminish the crimple. Table 1 shows how the light-emitting surface with different encapsulation techniques evolves with test time in the 85/85 test. We can clearly see from R3 that its light-emitting surface was free from dark spots after 760 h of testing, implying that the encapsulation film remained intact. Furthermore, the R3 works normally even when subjected to immersion under water, continuously driving at 6 V for 48 h (Figure 2b). Based on the Arrhenius–Peck equation (Equation (2)) [33], it can be simply estimated that a test with 760 h is equivalent to ~156,000 h working under normal operation condition at room temperature and RH of 50%.
(2)$$Acceleration\;factor=[{\left(\frac{Humidity{\;}_{high}}{Humidity{\;}_{low}}\right)}^{n}]*[e^{\left(\frac{Ea}{K}\right)*\left(\frac{1}{T1}-\frac{1}{T2}\right)}]$$
where *Humidity _low_* is the humidity level in normal operation conditions, *Humidity _high_* is the humidity level in the 85/85 test, *n* is the acceleration constant (*n* = 2.66 for the encapsulation of the metal cathode), *Ea* is the activation energy in electron-volts (eV), *K* is Boltzmann’s constant (8.62 × 10^−5^ eV/K), *T_1_* is the normal operation temperature in Kelvin, and *T_2_* is the temperature in the 85/85 test.

Although the *T_95_* operation lifetime of R4 was slightly better than that of R3, R4 had a poor storage lifespan under the 85/85 test. After all, Al is reactive to moisture and oxygen and the high-humidity and high-temperature conditions in the 85/85 test damaged the entire passivation structure. In short, the storage lifespan of R3 with the hybrid encapsulation of Si_x_N_y_/glass was nearly double than that of R1 using traditional desiccant/glass. Hybrid encapsulation is promising for lighting applications, which require working and storage in a harsh environment. For comparison, the 85/85 test was also carried out for devices that were not packaged and encapsulated with a glass cover without desiccant. The luminous surface is shown in Appendix A. The storage lifetime was only a few hours.

## 4. Conclusions

In this study, we have reported highly reliable hybrid encapsulation for OLEDs. Red OLEDs using Si_x_N_y_/glass and Si_x_N_y_/Al/glass as hybrid encapsulation exhibited *T_95_* (@1000 *cd/m*^2^) operation lifetime of over 76,000 and 84,000 h, respectively, which were 20% and 30% longer than that of OLEDs encapsulated by a typical desiccant/glass method. Most importantly, the red OLED encapsulated with Si_x_N_y_/glass realized a storage lifespan of over 17 years under ambient condition, 1.9 times longer than that encapsulated with a typical desiccant/glass. Hybrid encapsulation is promising for lighting applications, which require working and storage in a harsh environment.

## Figures and Tables

**Figure 1 molecules-27-02607-f001:**
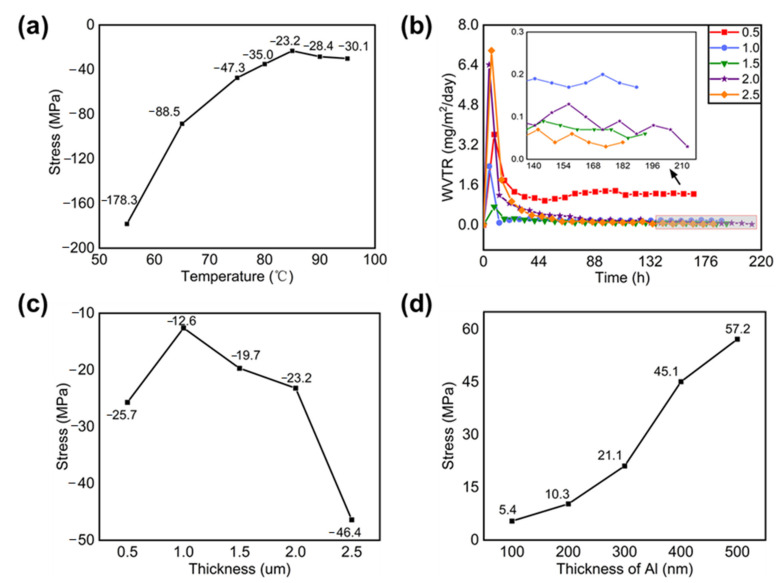
(**a**) Residual stress of 2-μm-thick Si_x_N_y_ at different deposition temperatures, (**b**) WVTR and (**c**) residual stress of the Si_x_N_y_ prepared at 85 °C with different deposition thicknesses, and (**d**) residual stress of Al with different deposition thicknesses.

**Figure 2 molecules-27-02607-f002:**
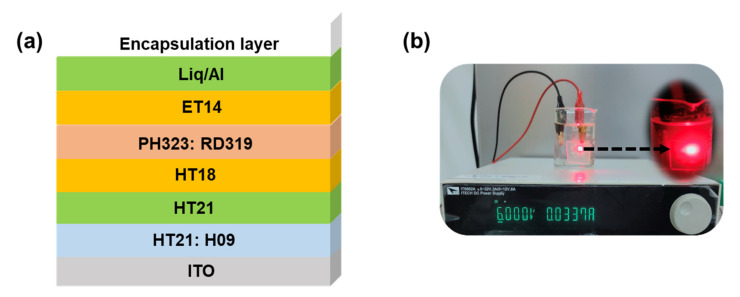
(**a**) Device structure in the present study and (**b**) a photograph showing Devices R3 subjected to water immersion for 48 h and continuously driving at 6 V.

**Figure 3 molecules-27-02607-f003:**
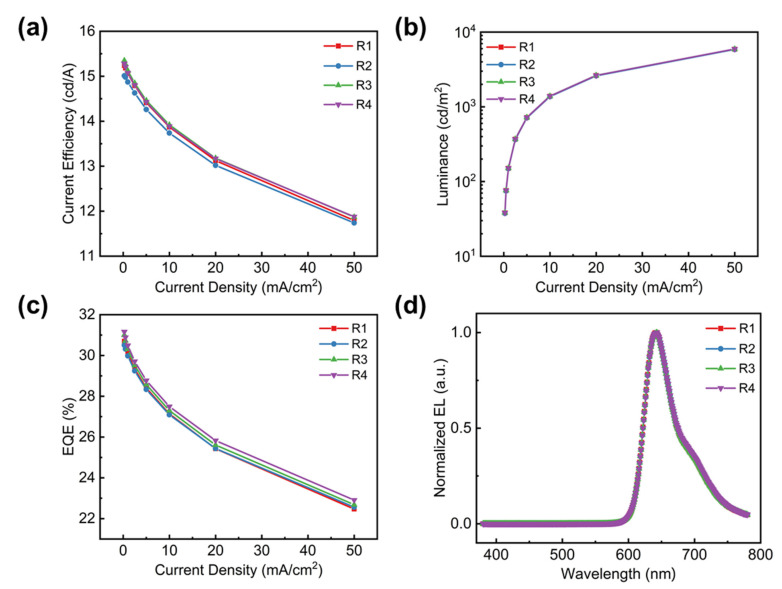
Device performance of devices R1–R4. (**a**) Current efficiency–current density, (**b**) luminance–current density, (**c**) external quantum efficiency–current density, and (**d**) EL spectra under a current density of 1 mA/cm^2^.

**Figure 4 molecules-27-02607-f004:**
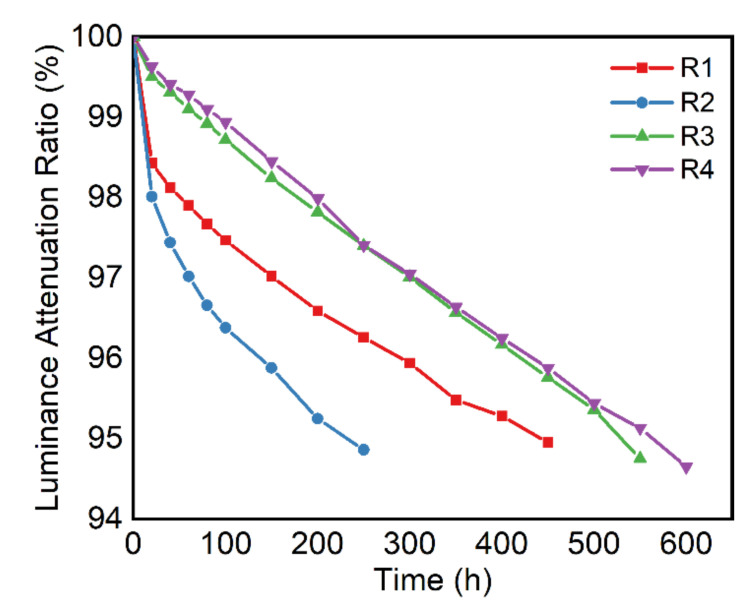
Operation lifetime of devices R1–R4 and *T_95_* at the current density of 50 mA/cm^2^.

**Table 1 molecules-27-02607-t001:** Luminous surface of devices R1–R3 under the 85/85 test.

Type	Fresh	5 h	20 h	50 h	200 h	400 h	600 h	800 h
R1						400 h		
R2				45 h				
R3								760 h

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
