# Peer review of "Hybrid Passivated Red Organic LEDs with Prolonged Operation and Storage Lifetime"

_molecules, 2022, doi:10.3390/molecules27092607_

Round 1

Reviewer 1 Report

The paper is dedicated to the investigation of organic light emitting diode encapsulation possibilities. Silicon nitride (SixNy) and Al was deposited by plasma enhanced chemical vapor deposition method. Deposited layer stress, water vapor transmission rate and OLED lifetime were measured. The work is interesting with the future use in the encapsulation of OLED.

Manuscript could be accepted in the journal “Molecules” after minor revision.

Questions:

  • Why WVTR is very high in first hours?
  • How is possible that the EQE of OLED is higher than 30%, if the planar structure limit is 5% for fluorescent and 20 % for phosphorescence dyes?

Author Response

Point 1: Why WVTR is very high in first hours?

Response 1: Many thanks for the reviewer’s question. The WVTR was very high in the first beginning because the sensors of the MOCON WVTR tester were in contact with air in prior to loading of our samples. After our samples were installed and the instrument was entirely enclosed, it took time for the sensors to stabilize and the WVTR then dropped to a steady state. The steady state reflects the actual permeation rate of water vapor penetrating into the thin film. We have added this description in the text for clarification.

Point 2: How is possible that the EQE of OLED is higher than 30%, if the planar structure limit is 5% for fluorescent and 20 % for phosphorescence dyes?

Response 2: Many thanks for the reviewer’s question. Nowadays, there have been lots of literatures reporting phosphorescent OLEDs with EQE over 20%. Theoretically, it is impossible. However, during the deposition process, it was proposed that the orientation of the emitter molecules will have a considerable influence on the out-coupling efficiency. Particularly, the horizontal oriented emitters will enhance the out-coupling of the generated light, thus increasing the EQE over 20%. However, the diploe orientation is not our focus in the present study. Also, we do not have such an equipment in our lab to prove this hypothesis. For more information, the referee can refer to these references:

Mayr et al., Organic Electronics, 15 (2014), p.3031.

Schmidt et al., Physical Review Applied 8 (2017), p.037001.

Reviewer 2 Report

The authors present an encapsulation method for OLED devices based on silicon nitride prepared by PECVD. The data shows that the method provides 30% longer storage lifetime compared to conventional encapsulation with bare glass. Under 85/85 test, the lifetime is almost doubled compared with bare glass encapsulation. The review has following comments:

  1. In Fig. 1(a), 85oC being the optimized temperature is not obvious. Error bars or more data points after 85oC are needed to confirm this. Some discussion about why the stress increases after 85oC would be helpful too.
  2. At what time are the WVTR values mentioned in line 105 and stress values in line 107-108 measured? The WVTR values are plotted against time in Fig. 1(b) but the time dependence is never discussed in the context.
  3. The data showing the low stress of SiN/Al composite mentioned in 119 should be provided.
  4. The discussion about R2 and R3 in line 158-162 should be before Fig. 3 is presented.
  5. The info in Table 1 is hard to identify. The dark spots analysis should be presented in some other form.
  6. The role of the Al layer is not convincing, since R4 has short lifetime in 85/85 time and the difference between R3 and R4 in Fig. 4 is too small. Without error bars, it is hard to tell if there is a difference at all.
  7. The lifetimes of devices with various SiN thicknesses should be tested. The stress and WVTR measurements in Fig.1 are only indirect evidences to the SiN performance as an encapsulation layer.
  8. Fabrication details of device R1 should be added since it is the main control device to compare with.

Overall, the results are interesting but not enough to demonstrate the advantage of the method. Comparison with other methods such as ALD processed AlOx should be included. Also some discussion on the fabrication cost would be helpful.

Author Response

Point 1: In Fig. 1(a), 85°C being the optimized temperature is not obvious. Error bars or more data points after 85°C are needed to confirm this. Some discussion about why the stress increases after 85°C would be helpful too.

Response 1: Many thanks for the reviewer’s advice. We have added more data (80°C and 90°C) in Figure 1. It is now clearer to see that the stress values between 80°C and 95°C were very similar. We cannot conclude that the stress increased after 85°C. There is a trade-off between the stress and substrate temperature. We all know that organic materials may suffer damage at a high temperature. Therefore, the optimal temperature in the present study was 85°C. We have added the corresponding description in the text.

Point 2: At what time are the WVTR values mentioned in line 105 and stress values in line 107-108 measured? The WVTR values are plotted against time in Fig. 1(b) but the time dependence is never discussed in the context.

Response 2: Many thanks for the reviewer’s comment. The stress values were measured after the films were freshly prepared. The WVTR values were obtained after 5 days of testing in MOCON instrument. When testing WVTR with the MOCON instrument, the WVTR was very high in the first beginning because the sensors of the MOCON WVTR tester were in contact with air in prior to loading of our samples. After our samples were installed and the instrument was entirely enclosed, it took time for the sensors to stabilize and the WVTR then dropped to a steady state. The steady state reflects the actual permeation rate of water vapor penetrating into the thin film. We have added this description in the text for clarification.

Point 3: The data showing the low stress of SiN/Al composite mentioned in 119 should be provided.

Response 3: Many thanks for the reviewer’s advice. We have added the data in Table S1.

Point 4: The discussion about R2 and R3 in line 158-162 should be before Fig. 3 is presented.

Response 4: Many thanks for the reviewer’s advice. Sorry for the misleading Fig. 3 is the device performance characteristics, from which we would like to point out that the encapsulation methods have no influence on the efficiencies and electroluminescence spectra. Then we started to discuss that the encapsulation merely affected the operation lifetime and storage lifespan. That’s why the discussion about R2 and R3 appeared in line 178-184 (corresponding to line 158-162 in the original manuscript), which is about the storage lifespan and the results are shown in Table 1.

Point 5: The info in Table 1 is hard to identify. The dark spots analysis should be presented in some other form. 

Response 5: Many thanks for the reviewer’s advice. The luminous surface of the devices as shown in Table 1 was observed under a microscope, and free of black spots. Once dark spots appeared on the device, the luminous surface changed significantly, as what can be seen in Table S2.

Point 6: The role of the Al layer is not convincing, since R4 has short lifetime in 85/85 time and the difference between R3 and R4 in Fig. 4 is too small. Without error bars, it is hard to tell if there is a difference at all.

Response 6: Many thanks for the reviewer’s question. R4 has similar operation lifetime as R3. The problem of R4 was merely unveiled in the 85/85 test since Al is reactive to moisture and oxygen, especially under the harsh conditions of high temperature and humidity in the test. That’s why R4 has a short storage lifespan. We have added the description in the text.

Point 7: The lifetimes of devices with various SiN thicknesses should be tested. The stress and WVTR measurements in Fig.1 are only indirect evidences to the SiN performance as an encapsulation layer.

Response 7: Many thanks for the important question. We did the lifetime test on 1um and 2um thick SiN and compared them with the device without any encapsulation. The T95 operation lifetime was 64, 160, and 240 hours for 0, 1 and 2 um SiN, respectively. We have added Figure S2 and description in the text.  

Point 8: Fabrication details of device R1 should be added since it is the main control device to compare with.

Response 8: Many thanks for the reviewer’s advice. The fabrication process of the device was added in the Experimental section.

Reviewer 3 Report

The article describes in detail the procedure of encapsulation of red organic light-emitting diodes (OLEDs) by low temperature thin film technology. The passivation of SixNy single-layer and SixNy/Al double-layer structures are manufactured by plasma enhanced chemical vapor deposition (PECVD) method. The passivation layers are characterized by the measurements of the residual mechanical stress and water vapor transmission rate. The authors compared performance parameters of encapsulated OLEDs manufactured by different methods. The external quantum efficiency (EQE) of OLEDs are calculated as a function of current density. The OLED encapsulated with SixNy/glass exhibited under ambient conditions the storage lifespan 1.9 times longer than that encapsulated with the commonly used (desiccant/glass) method, and the operation lifetime (T95) of 86 000 hours, which is 30% longer than with the conventional method. The results of high temperature/ high humidity tests are also presented. The results of the work are certainly a technological achievement and could be useful for scientific engineers working on OLEDs. The report is informative and clear, although the insight into the physical processes involved is not particularly deep. I recommend the Editor to publish this paper after some minor improvements.

  1. The molecular structures of used materials and fabrication details of investigated OLED architecture should be carefully depicted.
  2. The physical mechanisms responsible for EL quantum efficiency roll-off effect shown in Figure 3c should be mentioned (see eg. Materials 2020, 13, 1855).
  3. The abbreviation sccm needs to be explained.
  4. The term “current efficiency” in the caption of Figure 3c should be properly called current density.

Author Response

Point 1: The molecular structures of used materials and fabrication details of investigated OLED architecture should be carefully depicted.

Response 1: Many thanks for the reviewer’s advice. Since the present work is collaborated with a company, we were advised that the molecular structures cannot be disclosed. After all, the present study mainly focuses on the encapsulation method, and the hybrid encapsulation is universal for all devices, regardless of the molecular and device structures. Regarding the fabrication details, we have added the device fabrication process in the Experimental section. I hope the referee fully understands this situation.

Point 2: The physical mechanisms responsible for EL quantum efficiency roll-off effect shown in Figure 3c should be mentioned (see eg. Materials 2020, 13, 1855).

Response 2: Many thanks for the reviewer’s advice. The device we fabricated was based on phosphorescent emitter (we added this description in the experimental section). The dopant concentration was only 2%. The reduced efficiency roll-off (EQE only changed from ~31% to 26% at a high brightness of 3000 cd/m2) is attributed to the low dopant concentration (2% vol.%) used in the emitter, thus diminishing its concentration quenching and charge trapping effect in typical phosphorescent OLEDs. As the reviewer suggested, we already added this in the text to explain the efficiency roll off. Since this manuscript mainly focuses on device encapsulation, we did not spend too much to describe the EL mechanisms.

Point 3: The abbreviation sccm needs to be explained.

Response 3: Many thanks for the reviewer’s advice. The abbreviation sccm (standard cubic centimetres per minute) was added in the revised manuscript.

Point 4: The term “current efficiency” in the caption of Figure 3c should be properly called current density.

Response 4: Many thanks for the reviewer to point out the mistake. We have revised it.

Reviewer 4 Report

The authors Dan-Dan Feng , Shuang-Qiao Sun , Wei He , Jun Wang , Xiao-Bo Shi , Man-Keung Fung have submitted a manuscript entitled " Hybrid Passivated Red Organic LEDs with Prolonged Operation and Storage Lifetime" to the journal Moleculed of MDPI.

The introduction provides sufficient background and includes all relevant references. The research design is appropriate. The methods are adequately described. The results are clearly presented. Discussion of data and conclusions are adequately supported by the results.

English language and style are minor spell check required.

I do not detect plagiarism and I do not detect inappropriate citations.

In general, I do not see any ethical issues along the manuscript.

The topic of the manuscript is very interesting but I think that the manuscript can be improved if the authors can provide some major revisions to the manuscript.

These are my comments:

1) The authors at line 68 state "so as to further optimize the WVTR".

But the acronym WVTR is not specified in the previous paprt of the manuscript. I suggest to explain the acronym.

2) The authors at line 73 write "Commercially purchased ITO glass substrate". The acronym ITO should for indium tin oxide. The authors should write it. Moreover, ITO in tin doped indium oxide, but the doping with tin could be an important parameter, correlated with the final sheet esistance of the indium tin oxide substrates. Perhaps, the authors could specify these parameters of the indium tin oxide substrates.

3) The font size of the inset of Figure 1b is very small and for a reader it could be difficult to read the axis labels and legends. I syggest to increase the font size for the inset of Figure 1b.

4) Figure caption of Figure 2 is very short and it does not properly explain the twi figures. I suggest to the authors to explain the figure describing the left picture as Figure 2a and the right picture as Figure 2b. Moreover, all the materials employed in the light emitting diode structure in the left picture if Figure 2 should be explained in the figure caption.

5) Why in the equation 1 the T95 lifetime become T (that can be confused with transmission)? The authors could keep using T95 also in the equation.

6) It is not clear why the authors skip R4 encapsulation method from Table 1. In the table caption the authors state "Luminous surface of devices R1R4 under the 85/85 test", but R4 is missing.

7) An acceptable characterization of the light emitting diode is missing and it could be beneficial to have some spectroscopic information of the devices.

7.1) The electroluminescence spectrum of the light emitting diode should be reported. Moreover, could it be that the different encapsulation methods R1, R2, R3 and R4 play a role in changing the electroluminescence spectrum of the light emitting diode?

7.2) The authors could report the quantum efficiency of the light emitting diode. For example, in terms of photons emitted / injected charges. As for point 7.1, could it be that the different encapsulation methods R1, R2, R3 and R4 play a role in changing the quantum efficiency of the light emitting diode?

Author Response

Point 1: The authors at line 68 state "so as to further optimize the WVTR". But the acronym WVTR is not specified in the previous part of the manuscript. I suggest to explain the acronym.

Response 1: Many thanks for the reviewer’s advice. We have added the full term of WVTR (water vapor transmission rate).

Point 2: The authors at line 73 write "Commercially purchased ITO glass substrate". The acronym ITO should for indium tin oxide. The authors should write it. Moreover, ITO in tin doped indium oxide, but the doping with tin could be an important parameter, correlated with the final sheet resistance of the indium tin oxide substrates. Perhaps, the authors could specify these parameters of the indium tin oxide substrates.

Response 2: Many thanks for the reviewer’s advice and comment. We have added the full name of ITO and its parameters in the Experimental section.

Point 3: The font size of the inset of Figure 1b is very small and for a reader it could be difficult to read the axis labels and legends. I suggest to increase the font size for the inset of Figure 1b.

Response 3: Many thanks for the reviewer’s advice. We have revised the corresponding figure and tried to make it clear to reader.

Point 4: Figure caption of Figure 2 is very short and it does not properly explain the twi figures. I suggest to the authors to explain the figure describing the left picture as Figure 2a and the right picture as Figure 2b. Moreover, all the materials employed in the light emitting diode structure in the left picture if Figure 2 should be explained in the figure caption.

Response 4: Many thanks for the reviewer’s advice. We have separated Figure 2 into 2a and 2b. Since the present work is collaborated with a company, we were advised that the molecular structures cannot be disclosed. We have added the device fabrication process in the Experimental section (in red colour). I hope the referee fully understands this situation. After all, the present study mainly focuses on the encapsulation method, and the hybrid encapsulation is universal for all devices, regardless of the molecular and device structures.

Point 5: Why in the equation 1 the T95 lifetime become T (that can be confused with transmission)? The authors could keep using T95 also in the equation.

Response 5: Many thanks for the reviewer’s question. Sorry for confusing. We have changed T to T95.

Point 6: It is not clear why the authors skip R4 encapsulation method from Table 1. In the table caption the authors state "Luminous surface of devices R1–R4 under the 85/85 test", but R4 is missing.

Response 6: Many thanks for the reviewer’s question. We did not show R4 in Table 1 because R4 failed completely in the early beginning of the 85/85 test, although R4 has comparable operation lifetime with R3. The problem of R4 was merely unveiled in the 85/85 test since Al is reactive to moisture and oxygen, especially under the harsh conditions of high temperature and humidity in the test. That’s why R4 has a short storage lifespan. We have added the description in the text. Also, we have revised the caption in Table 1.

Point 7: An acceptable characterization of the light emitting diode is missing and it could be beneficial to have some spectroscopic information of the devices.

7.1) The electroluminescence spectrum of the light emitting diode should be reported. Moreover, could it be that the different encapsulation methods R1, R2, R3 and R4 play a role in changing the electroluminescence spectrum of the light emitting diode?

7.2) The authors could report the quantum efficiency of the light emitting diode. For example, in terms of photons emitted / injected charges. As for point 7.1, could it be that the different encapsulation methods R1, R2, R3 and R4 play a role in changing the quantum efficiency of the light emitting diode?

Response 7.1): Many thanks for the reviewer’s suggestion. We have added the EL spectra for Devices R1 to R4 (Figure 3). We can see that the encapsulation methods have no influence on EL.

Response 7.2): Many thanks for the reviewer’s question. We have the external quantum efficiency results of devices R1-R4, as can be seen in Figure 3c. The curves nearly coincided, demonstrating that different encapsulation methods have no effect on the quantum efficiency of our OLEDs.

Round 2

Reviewer 2 Report

The authors have answered most of the reviewer's questions and made changes to the manuscript accordingly. It is now suitable to be published in present form.

Reviewer 4 Report

The authors have provided a revised version of the manuscript "Hybrid Passivated Red Organic LEDs with Prolonged Operation and Storage Lifetime" for the journal Molecules.

In my opinion, the manuscript is improved with respect to the original submission. The authors have performed several revisions and integrations. 

The introduction provides sufficient background and includes all relevant references. The research design is appropriate. The methods are adequately described. The results are clearly presented. Discussion of data and conclusions are adequately supported by the results.

English language and style are minor spell check required.

I do not detect plagiarism and I do not detect inappropriate citations.

In general, I do not see any ethical issues along the manuscript.

In terms of originality, significance of content, quality of presentation, scientific soundness, interest to the readers, I think that the manuscript deserves publication in the journal Molecules. For this reason, I recommend the editorial board to accept the manuscript in the present form.